# Mechanisms of Resistance to KRAS^G12C^ Inhibitors

**DOI:** 10.3390/cancers13010151

**Published:** 2021-01-05

**Authors:** Victoria Dunnett-Kane, Pantelis Nicola, Fiona Blackhall, Colin Lindsay

**Affiliations:** 1Wythenshawe Hospital, Manchester University NHS Foundation Trust, Manchester M23 9LT, UK; victoria.dunnettkane@nhs.net; 2Department of Medical Oncology, The Christie NHS Foundation Trust, Wilmslow Road, Manchester M20 4BX, UK; pantelis.nicola@nhs.net (P.N.); fiona.blackhall@christie.nhs.uk (F.B.); 3Division of Molecular and Clinical Cancer Sciences, University of Manchester, Manchester M13 9PL, UK; 4Cancer Research UK Lung Cancer Centre of Excellence, Manchester M20 4BX, UK

**Keywords:** KRAS, G12C, oncogene, targeted therapy, NSCLC, drug resistance

## Abstract

**Simple Summary:**

KRAS is a gene that is commonly mutated in cancer, especially in pancreatic, lung and colorectal cancers. Despite its importance, scientists have for many years been unable to create a drug that can inhibit function of the KRAS protein. Recently, several drugs have been created that can inhibit the function of KRAS proteins that have a “G12C” mutation. There is hope that these G12C inhibitors will be useful for treating cancer in patients who have this mutation. In this review we discuss the problems that we may encounter with these drugs, especially the development of drug resistance to G12C inhibitors, and the approaches that we could use to overcome this.

**Abstract:**

KRAS is one of the most common human oncogenes, but concerted efforts to produce direct inhibitors have largely failed, earning KRAS the title of “undruggable”. Recent efforts to produce subtype specific inhibitors have been more successful, and several KRAS^G12C^ inhibitors have reached clinical trials, including adagrasib and sotorasib, which have shown early evidence of efficacy in patients. Lessons from other inhibitors of the RAS pathway suggest that the effect of these drugs will be limited in vivo by the development of drug resistance, and pre-clinical studies of G12C inhibitors have identified evidence of this. In this review we discuss the current evidence for G12C inhibitors, the mechanisms of resistance to G12C inhibitors and potential approaches to overcome them. We discuss possible targets of combination therapy, including SHP2, receptor tyrosine kinases, downstream effectors and PD1/PDL1, and review the ongoing clinical trials investigating these inhibitors.

## 1. Introduction

*RAS* is one of the most common human oncogenes, with gain-of-function mutations found in 27% of human cancers [1]. Three human *RAS* genes [neuroblastoma RAS viral (v-ras) oncogene homolog (*NRAS*), Harvey rat sarcoma viral oncogene homolog (*HRAS*), and Kirsten rat sarcoma viral oncogene homolog (*KRAS*)] code for four RAS GTPases: NRAS, HRAS, KRASa and KRASb, the latter two a result of alternative splicing.

RAS proteins cycle between an active GTP bound and an inactive GDP bound state. This process is tightly regulated by a diverse family of multi-domain proteins: guanine nucleotide exchange-factors (GEFs) and GTPase-activating proteins (GAPs). GEFs stimulate the dissociation of GDP and subsequent association of GTP, activating RAS proteins, while GAPs act to accelerate intrinsic GTP hydrolysis, converting RAS to its inactive state [2,3].

RAS proteins play a vital role in cellular proliferation, differentiation and survival, through their association with multiple downstream pathways, the best characterised of which are the Raf/Mek/Erk and PI3K/PTEN/Akt pathways [4]. Considering their role in driving hallmarks of cancer, and the frequency with which their mutations are identified in human cancer, RAS proteins and their downstream effectors have proved attractive targets for cancer therapy.

*KRAS* is overwhelmingly the most commonly mutated *RAS* isoform in cancer, comprising 85% of oncogenic *RAS* mutations [1]. *KRAS* mutations are most frequently found in epithelial cancers such as pancreatic, colorectal and lung adenocarcinomas [5,6]. According to data from TCGA PanCancer Atlas Studies, alterations in *KRAS* are found in around 64% of pancreatic tumours and 37% of colorectal tumours [5,6]. In non-small cell lung cancer (NSCLC), *KRAS* is mutated in around one third of patients, much more frequently than other oncogenic drivers e.g. *EGFR* (~15%), *ALK* rearrangements (~5%), *MET* mutations (~3%), for which targeted therapies are available [7].

Oncogenic mutations in the *RAS* family most commonly occur at codons 12, 13 and 61. The frequency of mutation at these sites varies between isoform and between malignancy types for a particular isoform. For *KRAS*, the vast majority (83%) of cancer-associated mutations occur at codon 12 [1]. Missense mutations at codons 12 and 13 are thought to limit interaction of GAP proteins with the GTPase site of RAS proteins, preventing their hydrolysis to an inactive state [8]. Five mutations (*G12D, G12V, G12C, G13D* and *Q61R*) account for 70% of all *RAS*-mutant patients [9].

Despite its appeal, KRAS has proven to be an elusive target. Efforts to produce GTP competitive inhibitors have been prevented by the pico-molar affinity with which RAS proteins bind GTP [10]. Furthermore, attempts to directly target KRAS have been hindered by its structure: KRAS presents a smooth surface with no deep hydrophobic pockets that would allow tight binding [11]. Consequently, past efforts have shifted focus to other targets [12]. RAS function depends on association with the plasma membrane, which is mediated through addition of C15 farnesyl isoprenoid lipid by farnesyltransferase [13]. This discovery led to the search for farnesyltransferase inhibitors (FTIs), which showed promising pre-clinical results [14,15]. Disappointingly, clinical trials failed to show anti-tumour activity [16,17]. More recent efforts investigating use of the FTI Tipifarnib in *HRAS*-mutant head and neck squamous cell carcinomas have shown promising early results [18,19]. Extensive research has also been channeled into the development of inhibitors that act downstream of RAS, but so far these have largely failed to show significant clinical benefits, either due to acquired resistance to the drug or toxicity limiting the maximum tolerated dose [20,21].

## 2. G12C Inhibitors

Recent efforts to produce mutant subtype specific KRAS inhibitors have been more fruitful, with the development of several small molecule inhibitors of KRAS^G12C^. *G12C* variants are most commonly found in non-small cell lung cancer, where they have been identified in around 11% of patients, as well as in 2.9% of all colorectal cancer (CRC) patients [22,23]. Despite the high prevalence of *KRAS* mutations in pancreatic adenocarcinoma, *G12C* is a minor subtype in this cohort (~1%) [22,24,25]. 

ARS-1620 was the first G12C specific inhibitor able to demonstrate in vivo efficacy, and since then several other related compounds with increased biological activity have been produced, the earliest of which to enter the clinic are adagrasib (MRTX849) and sotorasib (AMG-510) [26,27,28]. These compounds rely on mutant cysteine for binding, disrupting Switch-I/II and converting KRAS preference from GTP to GDP, thus holding KRAS in the inactive GDP bound state [29]. Several other molecules are in early clinical trials (Table 1). One of these trials using the allosteric inhibitor LY3499446 was discontinued early due to the development of unexpected toxicities and testing of JNJ-74699157 has also reportedly been discontinued. Despite these setbacks, phase 1 trials are ongoing for two novel inhibitors: GDC-6036 (Roche) and D-1553 (InventisBio) (NCT04449874, NCT04585035).

The fact that G12C inhibitors are able to bind KRAS^G12C^ in its inactive state can only be explained if the KRAS protein is not constitutively active in its GTP-bound state as previously assumed. In fact, KRAS^G12C^ appears to retain a near wild-type level of GTPase activity and undergoes nucleotide cycling in the cell [30,31]. G12C inhibitors act by preventing further nucleotide exchange, thus “trapping” the protein in a state of inactivity [26]. This exploitable property of KRAS^G12C^ appears unique: most KRAS proteins with mutations at codons 12, 13 and 61 have diminished GTPase activity compared to wild-type. An exception is KRAS^G13D^, which shows significantly elevated intrinsic exchange activity compared to wild-type RAS, and therefore may have a short-lived GDP-bound state [31]. The intrinsic GTPase activity of KRAS^G12C^ accounts not only for the efficacy of direct inactive-state inhibitors, but also widens the possibility of effective upstream targeting in *G12C*-mutant cancers.

Hallin et al. [28] extensively described the pre-clinical activity of the G12C inhibitor adagrasib (previously MRTX849). In a panel of 17 *KRAS^G12C^* mutant cancer cell lines, adagrasib was found to inhibit cell growth in most, but not all, cell lines, with IC50 values ranging between 0.01 μm and 0.973 μm in a 2D culture. This suggests a variable degree of sensitivity to the inhibitor, which was not accounted for by its consistent ability to bind and modify KRAS^G12C^. The cell lines with submaximal response showed only partial inhibition of ERK and S6 phosphorylation. In a panel of human *KRAS^G12C^* mutant cell-line and patient-derived xenograft models, adagrasib produced tumour regression >30% in 17/26 (65%) of samples. There was no significant correlation between in vitro response in viability assays and in vivo response. No genetic alteration, including *KRAS*-mutant allele frequency or *STK11* mutation status, was able to predict the anti-tumour activity, although baseline gene/protein expression of members of the HER family of receptor tyrosine kinase did exhibit a trend with the degree of anti-tumour response. A small molecule screen found that HER, SHP2, mTOR and CDK4/CDK6 inhibitors increased the response rate to MRTX849 in *KRAS^G12C^* mutant xenografts [28] *STK11* co-mutations in *KRAS* mutant NSCLC have been associated with poor response to therapy [32,33,34]; the fact that mutation in this gene did not correlate with drug resistance in cell lines offers some hope for this generally refractory group.

Similarly, Canon et al. described the pre-clinical activity of sotorasib (AMG-510). In cell viability assays, sotorasib impaired growth of all *KRAS^G12C^* lines except one, with IC50 values ranging from 0.004 μm to 0.032 μm. In immune competent mice, sotorasib successfully resulted in regression of *KRAS^G12C^* driven cancer, but this response lacked durability. This was partially overcome with a higher dose of sotorasib, which resulted in durable cures in 8/10 mice. Interestingly, AMG-510 was unable to produce a cure in mice that lacked T cells, with tumour growth rapidly recovering after a short response. Combination treatment with immune checkpoint (anti-PD1) inhibitors produced a sustained, complete response in most mice. Cured mice did not develop tumours when re-challenged with *KRAS^G12C^* cells [27]. 

A multicentre phase 1 trial examining the efficacy, safety and pharmacokinetics of sotorasib in 129 patients with advanced *KRAS^G12C^* mutant solid tumours has recently been reported [35]. This cohort consisted largely of patients with NSCLC and CRC (59 and 42 patients respectively), all heavily pre-treated. Just over half (56.6% of patients) experienced treatment-related adverse effects of any grade, and 15 patients (11.6%) experienced grade 3 or 4 treatment related adverse events; the most common of which were deranged liver function tests, diarrhoea and anaemia. In the NSCLC cohort, 32% (19/59) of patients had a confirmed partial response, and 88% (52/59) showed disease control, leading to a median progression free survival of 6.3 months (compared to around 2.9 months in patients with NSCLC receiving standard 2nd or 3rd line chemotherapy [36]). In the CRC cohort, 7.1% (3/42) demonstrated a partial response to treatment, and 73.8% (31/42) had disease control, leading to a median progression free survival of 4.0 months [35].

Adagrasib is currently being evaluated in a phase 1/2 trial in patients with *KRAS^G12C^* mutant advanced or metastatic solid tumours who have previously been pre-treated with chemotherapy or anti-PD-1/PD-L1 therapy (NCT03785249, Table 1). Almost in parallel with sotorasib, an update on this trial has recently been presented, revealing that 79 patients with pre-treated NSCLC have so far received adagrasib. The most commonly reported treatment-related adverse effects included nausea (54%), diarrhoea (48%), vomiting (34%), fatigue (28%) and raised ALT (23%). Fifty-one patients were evaluable for clinical activity and of these 45% (23/51) had objective response and 96% (49/51) had disease control [37]. Adagrasib had also been given to 22 patients with pre-treated CRC; in this group 17% (3/18) of evaluable patients had a confirmed objective response, and 94% (17/18) patients exhibited disease control [38].

Clearly these studies show great promise of efficacy, even in subgroups that have generally poor response to therapy, such as patients with *KRAS/STK11* co-mutation. However, the variable response to treatment seen in pre-clinical models and patients, and the evidence of disease relapse in murine models, points towards drug resistance to KRAS^G12C^ inhibitors. This is unsurprising and has been well documented with other targeted cancer therapies. Understanding the mechanisms of intrinsic and acquired resistance to KRAS^G12C^ inhibitors is therefore vital for maximising their therapeutic potential [39]. 

## 3. Mechanisms of Resistance to KRAS^G12C^ Inhibitors

A lack of dependency on KRAS signalling could account for some of the intrinsic resistance noted in pre-clinical models, which may underlie the variable response seen in patients [27,39,40]. RAS proteins exert their effects through multiple pathways, including the MAPK/ERK and PI3K/AKT/mTORC1 pathways, although the PI3K pathway is not dependent on RAS alone for activation [41]. It has been shown that *KRAS*-mutant lung and pancreatic ductal adenocarcinoma (PDAC) cell lines vary in their dependence on RAS signalling [42]. Even with complete KRAS inhibition, a subset of *KRAS*-mutant PDAC cells remain viable, and a significant majority of these demonstrate PI3K-dependent MAPK signalling and sensitivity to MAPK pathway inhibitors [43]. Other studies have identified amplification of the transcriptional coactivator YAP1 as a means of bypassing KRAS inhibition in cancer cell lines [44,45]. 

The phenomenon of acquired resistance to targeted therapy has frequently been noted with downstream inhibitors of the RAS pathway, examples of which are too numerous to comprehensively cover here but have been reviewed elsewhere [10,12,46]. Clinical attempts to target the MAPK pathway have previously focused on MEK, but disappointingly these inhibitors have proven to have limited therapeutic activity in *KRAS*-mutant lung cancer patients [47]. The large multinational randomised control trial SELECT-1 compared docetaxel alone or in combination with the MEK inhibitor selumetinib in 510 patients with *KRAS*-mutant NSCLC, finding no significant difference in progression-free or overall survival [48]. Similarly, a phase 2 study of the MEK inhibitor trametinib found no survival benefit over docetaxel in pre-treated *KRAS*-mutant NSCLC patients [49]. 

This is likely explained in part by the activation of alternative RAS dependent pathways, but it was also noted that MEK inhibition leads to relief of negative feedback mechanisms and subsequent upregulation of upstream receptor tyrosine kinases (RTKs) [50,51]. A similar process is seen with targeted BRAF inhibition in *BRAF^V600E^* mutant cancers, where relief of negative feedback mechanisms leads to EGFR-mediated activation of RAS and CRAF. This is particularly apparent in *BRAF^V600E^* mutant colorectal cancers, which express higher levels of EGFR than melanoma cells [52,53]. Interestingly, loss of wild-type *KRAS* has been shown to increase sensitivity to MEK inhibitors in KRAS-mutant cell lines [54]. The mechanism by which wild-type KRAS promotes resistance to MEK inhibition has not been fully elucidated, but appears to rely on its ability to dimerise with mutant KRAS [55]. Dimerisation is also required for the function of oncogenic mutant KRAS. Disruption of dimerisation between wild-type and mutant KRAS may not only increase the sensitivity to MEK inhibitors, but may also inhibit the biological function of mutant KRAS [55].

Acquired resistance to G12C inhibitors has already been documented. *KRAS^G12C^* mutant NSCLC cell lines treated with ARS-1620 showed variable reactivation of the MAPK pathway. As seen with downstream inhibitors, blockade of RTK overcame resistance in a subset of models, however, PI3K pathway inhibition appeared to be more broadly effective [56]. A recent study by Xue et al. [57] identified a heterogenous response of *KRAS^G12C^* mutant cancer cells to G12C inhibition with ARS-1620. Distinct subgroups with different drug-response trajectories were identified. While most cells entered a quiescent (G0) state, others quickly regained RAS activity and resumed proliferation. This response seemed to be mediated by the production of new KRAS^G12C^ in response to reduced MAPK signaling. The new KRAS^G12C^ was maintained in the active GTP bound, drug insensitive state by epidermal growth factor receptor (EGFR) and phosphatase SHP2 signaling. Aurora kinase A (AURKA) was also identified as necessary for escape from drug-induced quiescence, which may be due to a stabilizing reaction between AURKA, KRAS^G12C^ and the downstream effector CRAF [57].

In contrast to the theory of novel KRAS^G12C^ production maintaining cell proliferation, a study by Ryan et al. found that adaptive RAS pathway reactivation following KRAS^G12C^ inhibition was mediated through wild-type RAS. In line with previous work, it was identified that rapid feedback reactivation was driven by multiple RTKs, which they found to induce HRAS and NRAS activity and subsequent reactivation of KRAS^G12C^ independent signaling. Furthermore, this study identified no single RTK that was critical for all KRAS^G12C^ models but found that co-inhibition of SHP2 resulted in universal and sustained inhibition of feedback reactivation [58].

Despite similar study designs, the differing conclusions of the two aforementioned studies raises the question of whether resistance is mediated through wild-type RAS, newly produced mutant KRAS or both [39,57,58]. Regardless, both studies identified the importance of RTK and SHP2 activation in acquired resistance. SHP2 is a phosphatase that mediates signaling from multiple RTKs to the RAS pathway, and as such offers an attractive target for KRAS^G12C^ combination therapy [59]. Mirati are testing this combination in a phase 1/2 clinical trial (NCT04330664) and several other trials of SHP2 inhibitors alone or in combination with other agents are currently underway (Table 1). 

## 4. Maximising the Potential of G12C Inhibitors 

It is clear that intrinsic and acquired mechanisms of resistance will need to be overcome in order to maximise the therapeutic potential of KRAS^G12C^ inhibitors. Due to the specificity of G12C inhibitors, they should provide a better therapeutic window than downstream RAS pathway inhibitors, and co-inhibition with other targeted agents may be better tolerated in patients (Figure 1). Future studies will need to confirm the expected tolerable toxicity profiles of combination therapy. 

### 4.1. Upstream Co-Inhibition

As mentioned previously, SHP2 is a central node in RTK and RAS signalling, and has been postulated as a valuable co-inhibitory target in KRAS^G12C^ signalling. Inhibition of SHP2 has been investigated with other inhibitors of the RAS/MAPK pathway and shown to overcome RTK driven acquired resistance to MEK inhibition [60,61]. SHP2 inhibition has also been demonstrated to have single agent efficacy in pre-clinical models with oncogenic mutations in the RAS/RAF/MEK/ERK pathway [62]. Both Xue et al. and Ryan et al. identified SHP2 as a vital mediator of acquired resistance to KRAS^G12C^ inhibition [57,58]. Results from a phase 1 clinical trial using RMC-4630, a selective SHP2 inhibitor, show some early evidence of efficacy in NSCLC patients harbouring a *KRAS^G12C^* mutation, with 71% (5/7) of patients showing disease control [63]. RMC-4630 is also being tested in combination with combimetinib, a MEK inhibitor, and a recent update has shown some preliminary evidence of anti-tumour activity in *KRAS*-mutant colorectal cancer with tumour reduction in 37.5% of patients (3/8) [64]. 

A recently published study by Fedele et al. found that SHP2 inhibitors increase KRAS-GDP occupancy, increasing the effect of G12C inhibitors in vitro. They also identified that SHP2 inhibition was able to overcome G12C inhibitor resistance in vitro and in PDAC and NSCLC models. Either inhibitor alone had minimal effects on murine PDAC and NSCLC models, whereas G12C/SHP2 co-inhibition conferred extended survival in all models with no evidence of toxicity. SHP2 co-inhibition also appears to lead to a more favourable immune microenvironment, and sensitised tumours to PD-1 inhibition [65]. In agreement with these findings, the SHP2 inhibitor TNO155 has been shown to overcome feedback reactivation of RTK signalling induced by G12C inhibition and was found to act synergistically with the G12C inhibitor Cpd 12a to inhibit proliferation in *KRAS^G12C^* cell lines. This synergistic effect was the most significant of all TNO155 combinations tested in the study and was also postulated to be secondary to increased GDP occupancy of KRAS [66]. Taken together, this evidence suggests that SHP2 may prove to be a useful target to overcome KRAS^G12C^ inhibitor resistance. Indeed, both sotorasib and adagrasib will be tested in combination with SHP2 inhibitors in early-phase clinical trials [67] (Table 1).

Another potential approach to overcome KRAS^G12C^ inhibitor resistance is to target RTKs (Figure 1), the activity of which is upregulated by G12C inhibition [39,56,58]. Lito et al. demonstrated that inhibition of MET, SRC or FGFR in combination with G12C inhibition potentiated growth inhibition in two *KRAS^G12C^* mutant cell lines, whereas concurrent EGFR inhibition potentiated growth inhibition in two different *KRAS^G12C^*-mutant cell lines, showing a heterogenous response between them [30]. In agreement, Ryan et al. demonstrated that numerous RTKs are involved in the adaptive feedback mechanism to G12C inhibition, and that the pattern of RTK dependence varies between *KRAS^G12C^* mutant cancers. Afatinib (a pan-HER inhibitor) and BGJ398 (an FGFR inhibitor) showed efficacy in combination with G12C inhibition in many, but not all models. [58] Another study found that, while RTK inhibitor combinations produced some of the strongest synergistic effects, this was not consistently seen across cell models [56]. This data suggests that targeting a specific RTK is unlikely to be universally effective across multiple *KRAS^G12C^* mutant cancers. Taken together, the results described so far paint a variable pattern of resistance which will require further definition and may ultimately influence patient selection criteria at the point of resistance.

One clue towards understanding this discrepancy may come from the relatively poor response to G12C inhibitors in colorectal cancer lines. As was recently reported, objective response rate to sotorasib was much poorer in CRC patients (7.1%) than in those with NSCLC (32%) [35]. In an effort to understand this difference it has been shown that, in comparison to NSCLC lines, CRC cell lines failed to demonstrate sustained ERK inhibition after treatment with sotorasib, and that *G12C*-mutant CRC cell lines retained their dependency on EGFR for downstream signalling [68]. Furthermore, using *KRAS^G12C^* mutant CRC patient-derived xenografts, they found that the combination of sotorasib and cetuximab, an EGFR monoclonal antibody, produced profound tumour regression which was absent with either agent alone. The authors proposed that, as CRC is dominated by wild-type EGFR signaling, combination therapy with anti-EGFR mAbs will be more effective than EGFR TKIs. Certainly, considering the limited response to single agent G12C inhibitors in CRC, combination therapy will be vital, and trials combining G12C inhibitors with anti-EGFR mAbs are underway (Table 1). 

### 4.2. Downstream Co-Inhibition 

KRAS activates multiple downstream pathways, and consequently inhibition of downstream targets would be expected to offer less efficacy than direct or upstream targeting [39]. This has proven largely true, although Misale et al. identified that inhibition of PI3K in combination with G12C inhibition consistently increased efficacy across multiple cell lines [56]. The authors postulated that this effect may due to either PI3K-mediated reactivation of ERK signaling through adaptor proteins, or due to the combined shutdown of two major cell signaling pathways. The importance of this relationship is supported by studies showing a synergistic effect between MEK inhibitors and PI3K inhibitors in lung and PDAC models [69,70]. However, the increased efficacy appears to come at the expense of greater toxicity, which may limit the usefulness of this combination therapy in the clinic [71,72]. 

In the context of disappointing clinical results from inhibition of downstream targets, one elegant study using advanced RAS preclinical models suggested that effective combination therapy may be sought by targeting both upstream and downstream targets. Molina-Arcus et al. [73] found that co-targeting mTOR/IGF1R and either MEK or KRAS^G12C^ showed efficacy in mouse models of *KRAS*-mutant lung cancer, with less toxicity noted in the G12C inhibitor combination. However, further clinical studies with this triple combination therapy are necessary to assess the toxicity profile in patients.

### 4.3. Cell Cycle Checkpoint Co-Inhibition 

Several clinical trials currently underway have incorporated the addition of a CDK4/6 inhibitor (Table 1, Figure 1). CDK4 and CKD6 cooperate with D-type cyclins, leading to progression through the G1/S cell cycle checkpoint. Targeting CDK would theoretically increase the effects of G12C inhibitors by maximising cell cycle arrest, and combination with the CDK4/6 inhibitor palbociclib has been shown to have synergistic effects with ARS-1620 and adagrasib [28,74].

Metabolic reprogramming is an important hallmark of cancer cells and presents a potential method for targeting *KRAS* driven malignancies. In order to meet increased requirements for cell growth, cancer cells redirect intermediates from the tricarboxylic acid (TCA) cycle, which need to be replenished with the amino acid glutamine (Gln). The reliance on Gln creates a Gln dependent G1 cell cycle checkpoint, which *KRAS* mutant cancer cells are able to bypass in states of Gln depletion, instead arresting in S/G2 [75]. This can be exploited by utilising methods to induce Gln depletion, while simultaneously employing G2/S specific cytotoxic drugs [76]. This approach proves that metabolic checkpoints can be used in synthetic lethality screens. This approach could be used to overcome resistance to G12C inhibitors, which are known to sequester cells into a quiescent state, while adapting cell progress through G1/S and G2 [57]. 

Glutamine metabolism has also been suggested as a potential therapeutic target in *KRAS* mutant PDAC. In this context, mutant KRAS upregulates the transcription factor NRF2 which confers chemoresistance by altering pathways regulated by Gln metabolism. Glutamine depletion and use of glutaminase inhibitors sensitises PDAC cells to the chemotherapeutic agent gemcitabine [77]. Altered metabolic pathways offer another possible target for combination therapy in resistant tumours.

### 4.4. Immune Checkpoint Co-Inhibition 

Canon et al. [27] found that AMG-510 led to a pro-inflammatory tumour microenvironment and acted synergistically with immune checkpoint inhibitors. In their study, AMG-510 produced durable tumour response in immune competent mice, but not in mice that lacked T cells, suggesting that anti-tumour T cell activity may be required for a sustained response. Combined sotorasib with PD-1 inhibition increased CD8+ T cell infiltration into the tumour and conferred a durable complete response in most mice [27]. Fedele et al. found that G12C/SHP2 inhibition increased CD8+ T cell infiltration into the tumour microenvironment, and triple therapy with PD-1 inhibitors resulted in greater tumour regression in PDAC murine models [60]. This provides a rationale for the combination of KRAS^G12C^ inhibitors with immunotherapy, and clinical trials combining sotorasib/adagrasib with anti-PD1/L1 are planned (NCT03600883, NCT03785249, clinicaltrials.gov). The importance of this is underlined by the current anchoring of lung cancer systemic treatment by checkpoint inhibitors: it is unlikely that G12C inhibitors will succeed in first-line treatment without successful progress of this combination.

## 5. Conclusions

In conclusion, KRAS^G12C^ specific inhibitors are a testament to the progress that has been made in targeting an oncogene that was previously considered “undruggable”. Two KRAS^G12C^ inhibitors, adagrasib and sotorasib, have shown efficacy in early clinical trials and further clinical studies are underway [27,40]. Like the downstream targeted therapies that have preceded them, it is clear that G12C inhibitors will induce acquired resistance. It remains unclear whether this resistance is mediated through wild-type RAS [58], new KRAS^G12C^ production, or other mechanisms [57]. However, studies agree that reactivation of RTK-SHP2 signalling may go some way to explaining this adaptive response. Finally, although adaptive signalling pathways appear important in experimental models, it is unclear what the relevance of this will be in patients [78]. 

It seems likely that drug combinations will be required to achieve maximum responses, and due to the specificity of G12C inhibitors, there is optimism that these drug combinations will be tolerated. Indeed, recently updated phase 1 results confirm that sotorasib and adagrasib are well tolerated in patients, with 11.6% of patients receiving sotorasib experiencing grade 3/4 treatment-related adverse effects [35]. Several possible inhibitory partners have been identified, including SHP2, receptor tyrosine kinases, PI3K and PD-1/L1 [27,56,65]. However, there remains a lack of consensus on which of these targets should be prioritised. As these combinations move through clinical trials, it will be important to investigate the patterns and mechanisms of resistance induced in patients, using this information to define patient selection beyond KRAS^G12C^ where necessary.

## Figures and Tables

**Figure 1 cancers-13-00151-f001:**
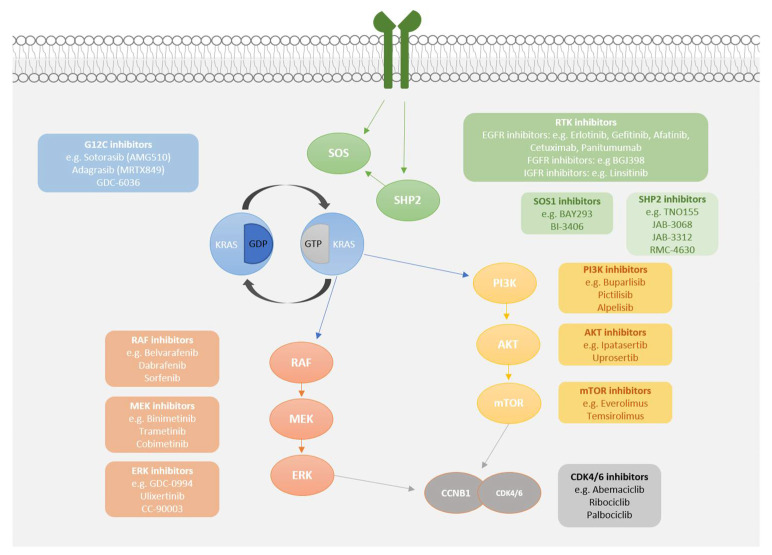
KRAS^G12C^ potential co-inhibitory targets.

**Table 1 cancers-13-00151-t001:** Registered trials of G12C inhibitors or SHP2 inhibitors on clinicaltrials.gov.

Clinicaltrials.gov Reference	Drug Name	Target	Stage of Development	Estimated Enrolment	Design
NCT03600883 CodeBreak 100	Sotorasib (AMG 510) +/− anti PD-1/L1 (Amgen)	KRAS^G12C^	Phase 1/2 recruiting	533	AMG 510 monotherapy in patients with advanced solid tumours + KRAS^G12C^ mutation and AMG 510 combination therapy (anti-PD1/L1) in patients with advanced NSCLC + KRAS p.G12C mutation
NCT04185883 CodeBreak 101	Sotorasib (AMG 510) (Amgen)	KRAS^G12C^	Phase 1b recruiting	456	AMG 510 monotherapy and in combination with other anti-cancer therapies in patients with advanced KRAS^G12C^ mutant solid tumours
NCT04380753 CodeBreak 105	Sotorasib (AMG 510) (Amgen)	KRAS^G12C^	Phase 1 recruiting	12	AMG 510 in patients of Chinese descent with advanced/metastatic solid tumours with KRAS^G12C^ mutation
NCT04303780 CodeBreak 200	Sotorasib (AMG 510) (Amgen) vs. docetaxel	KRAS^G12C^	Phase 3 recruiting	650	AMG 510 vs docetaxel in pre-treated locally advanced and unresectable or metastatic NSCLC patients with KRAS.^G12C^ mutation
NCT03785249	Adagrasib (MRTX849) +/− pembolizumab/ cetuximab/ afatinib (mirati therapeutics)	KRAS^G12C^ (+/− anti-PD-1, EGFR)	Phase 1/2 recruiting	391	MRTX849 in patients with advanced solid tumours with a KRAS^G12C^ mutation
NCT04330664	Adagrasib (MRTX849), TNO155 (Mirati therapeutics and Novaratis)	KRAS^G12C^, SHP2	Phase 1/2 recruiting	148	Combination of MRTX849 with TN0155 in patients with advanced solid tumours and KRAS^G12C^ mutation
NCT04449874	GDC-6036 (Roche)	KRAS^G12C^	Phase 1/2 recruiting	108	GDC-6036 in patients with advanced or metastatic solid tumours with a KRAS^G12C^ mutation
NCT04165031	LY3499446 (Eli Lilly) +/− Abemaciclib /cetuximab/erlotinib vs docetaxel	KRAS^G12C^ (+/− CDK4/6, EGFR)	Phase 1: terminated due to toxicity	230	LY3499446 in patients with advanced solid tumours and KRAS^G12C^ mutation
NCT04006301	JNJ-74699157 (Wellspring biosciences and Janssen)	KRAS^G12C^	Phase 1: terminated due to toxicity	10 (actual)	Complete: JNJ-74699157 in patients with advanced solid tumours (including NSCLC, CRC) with a KRAS^G12C^ mutation
NCT04585035	D-1553 (InventisBio)	KRAS^G12C^	Phase 1/2 not yet recruiting	200	D-1553 in patients with advanced or metastatic solid tumours with KRAS^G12C^ mutation
NCT03114319	TNO155 +/− EGF816 (Novaratis)	SHP2 (+/−EGFR)	Phase 1 recruiting	255	TNO155 alone or in combination with EGF816 in patients with advanced selected solid tumours (EGFR or KRAS^G12C^ mutant NSCLC, oesophageal SCC, HNSCC, melanoma)
NCT04000529	TNO155 + spartalizumab/ribociclib (Novaratis)	SHP2 (+ anti-PD-1/ CDK4/6)	Phase 1b recruiting	126	TNO155 in combination with spartalizumab or ribociclib in selected malignancies (NSCLC, CRC, HNSCC, GIST, oesophageal SCC)
NCT03565003	JAB-3068 (Jacobio Pharmaceuticals)	SHP2	Phase 1/2 recruiting	120	JAB-3068 in patients with advanced solid tumours in China
NCT03518554	JAB-3068 (Jacobio Pharmaceuticals)	SHP2	Phase 1 recruiting	45	JAB-3068 in patients with advanced solid tumours
NCT04121286	JAB-3312 (Jacobio Pharmaceuticals)	SHP2	Phase 1 recruiting	24	JAB-3312 in patients with advanced solid tumours in China
NCT04045496	JAB-3312 (Jacobio Pharmaceuticals)	SHP2	Phase 1 recruiting	24	JAB-3312 in adult patients with advanced solid tumours
NCT03989115	RMC-4630 + cobimetinib/osimertinib (Revolution Medicines and Sanofi)	SHP2 (+ MEK/EGFR)	Phase 1/2 recruiting	168	RMC-4630 + cobimetinib in patients with relapsed/refractory solid tumours and RMC-4630 + osimertinib in patients with EGFR mutant, locally advanced or metastatic NSCLC
NCT03634982	RMC-4630 (Revolution Medicines and Sanofi)	SHP2	Phase 1 recruiting	210	RMC-4630 monotherapy in patients with relapsed/refractory solid tumours
NCT04252339	RLY-1971 (Relay Therapeutics)	SHP2	Phase 1 recruiting	52	RLY-1971 in patients with advanced or metastatic solid tumours
NCT04528836	BBP-398 (Navire Pharma)	SHP2	Phase 1 not yet recruiting	60	BBP-398 in patients with MAPK pathway- or RTK-driven advanced solid tumours

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
