# Peer review of "Mechanisms of Resistance to KRASG12C Inhibitors"

_cancers, 2021, doi:10.3390/cancers13010151_

Round 1

Reviewer 1 Report

This work is the interesting and comprehensive summary for the understanding of the treated topic. The abstract provides adequate information in a few lines on the discussion of the following paragraphs. The introduction is comprehensive and offers an adequate background of knowledge on the role of RAS family. The role of the treated inhibitors and the limitations in the single use of such agents is clear. The importance of the co-inhibition upstream of the RAS pathway to overcome the resistance to KRASG12C inhibitors is quite clear. The references given are exhaustive for further explanations during the discussion of the topic.

The manuscripts titled "Mechanisms of resistance to KRASG12C inhibitors"by Dunnet-Kane et al discusses the current evidence for the KRASG12C inhibitors, the mechanisms of the intrinsic and acquired resistance to G12C inhibitors and the potential approaches to overcome them.It summarizes ongoing clinical trials studying these inhibitors. Furthermore, it describes the combination therapy and its possible targets. Among the most treated inhibitors are described ARS-1620 (first known inhibitors),Adagrasib and Sotorasib that they are the specific inhibitors for KRAS protein with the mutant cysteine. These compounds have shown efficacy in early clinical trials but they are limited in vivo by the development of drug resistance. The authors of this manuscripts reported two studiesexplaining that the acquired resistance is mediated by wild-type RAS, the newly produced mutant KRASG12C or both. But, these two studies still remain to be clarified. However,theboth studies has recognized the importance of the RTK and SHP2 activation in acquired resistance. Therefore, RTK and SHP2 can be two possible upstream targets in combination with KRASG12C inhibitors.The co-inhibition can be useful to increase the effects of KRASG12C inhibitors. It is reported that the downstream co-inhibition offers less efficacy than direct or upstream targeting. But, there are studies showing that the combination of ARS-1620 and Adagrasib with the CDK4/6 inhibitors have synergistic effects on the cell cycle arrest. Finally, it is reported that combined Sotosarib with PD-1 inhibition increased the CD8+ T cell infiltration into the tumor and subsequent tumor regression.

This work is the interesting and comprehensive summaryfor the understanding of the treated topic.The abstract provides adequate information in a few lines on the discussion of the following paragraphs.The introduction is comprehensive and offers an adequate background of knowledge on the role of RAS family. The role of the treated inhibitors and the limitations in the single use of such agents is clear.The importance of the co-inhibition upstream of the RAS pathway to overcome the resistance to KRASG12C inhibitors is quite clear.The references given are exhaustive for further explanations during the discussion of the topic.

Minor revisions.

  • In paragraph "G12C inhibitors" it is suggested to report the cell viability assays graphs for a faster understanding of the effects of the Adagrasib and Sotorasib on growth cell lines.

  • Re-reading of the manuscript is suggested to avoid errors. For example in "G12C inhibitors" paragraph the G12C inhibitor is not ARS-160 but ARS-1620.

Author Response

Thank you for the kind and thorough review of our manuscript. We appreciate the comments that you’ve provided and have acted on them as below.

In paragraph "G12C inhibitors" it is suggested to report the cell viability assays graphs for a faster understanding of the effects of the Adagrasib and Sotorasib on growth cell lines”

Descriptions of the graphs have now been provided in section 2, starting on lines 111 and 129 as follows “In a panel of 17 KRASG12C mutant cancer cell lines, adagrasib was found to inhibit cell growth in most, but not all, cell lines, with IC50 values ranging between 0.01μm and 0.973μm in a 2D culture” and  “In cell viability assays sotorasib impaired growth of all KRASG12C lines except one, with IC50 values ranging from 0.004μm to 0.032μm”

“Re-reading of the manuscript is suggested to avoid errors. For example in "G12C inhibitors" paragraph the G12C inhibitor is not ARS-160 but ARS-1620”

Thank you for pointing out this error. We have re-read and amended the manuscript as needed.

Reviewer 2 Report

Timely review article by Dr. Lindsey and group on mechanistic view of resistance created by KRAS G12C inhibitors. Few things need to be addressed before it is ready for acceptance. They are as follows:

  1. Reference 65, has been accepted in JEM journal. Please update it.
  2.  Page 6 line 318, reference 60 is wrong. It will be reference 65 instead.
  3. Ongoing studies of one of the KRAS G12C inhibitors, MRTX849, have indicated NRF2’s involvement in MRTX849 resistance (doi:10.1038/s43018-019-0016-8 (2020). While (DOI: 10.1158/0008-5472.CAN-19-1363) it has been shown that NRF2 is responsible for creating chemoresistance in KRAS mutant pancreatic cancers. This study also suggests role of NRF2 mediated altered metabolism and stress granules for the same. Authors should add few lines on this topic as another possible approach for further research.
  4. Following up with above mentioned topic, it is also worthwhile mention about how to exploit metabolic checkpoints to break the resistance of KRAS G12C inhibitors. It has been mentioned in this manuscript elaborating the study from Lito's group stating that cell cycle arrest upon KRAS G12C inhibitors treatment. Lito's group has shown that treatment with G12Ci induced a quiescent state (G0) that was transcriptionally distinct from G1. While it has been shown (DOI:10.4161/23723548.2014.963445) that how to utilize metabolic checkpoints to create synthetic lethal phenotype in KRAS-mutants. Authors should add few lines on this aspect by referring mentioned works above. This will add a new aspect in this review which will help readers to think in broader aspect.
  5. in case of SHP2+KRAS G12C combination study, another new work came out from Novartis group (DOI: 10.1158/1078-0432.CCR-20-2718). It has been shown that SHP2 inhibitor TNO155 is efficient in KRAS G12C cancer cells: it efficiently blocked the feedback activation of wild-type KRAS or other RAS isoforms induced by G12C inhibitors and efficiently enhanced the efficacy of G12C targeted therapy.  

Authors should edit the text accordingly after incorporating the above mentioned references and topics. It will be helpful for readers to understand the topic of this manuscript in global manner. 

Author Response

Thank you for your very helpful and insightful comments on our manuscript which we have acted on and amended as below.

  1. Reference 65, has been accepted in JEM journal. Please update it.
  2. Page 6 line 318, reference 60 is wrong. It will be reference 65 instead.

Thank you for drawing our attention to both of these. We have amended reference 65 and removed the statement “awaiting peer review” in the text. The incorrect reference on page 6 has been changed to reference 65.

  1. Ongoing studies of one of the KRAS G12C inhibitors, MRTX849, have indicated NRF2’s involvement in MRTX849 resistance (doi:10.1038/s43018-019-0016-8 (2020). While (DOI: 10.1158/0008-5472.CAN-19-1363) it has been shown that NRF2 is responsible for creating chemoresistance in KRAS mutant pancreatic cancers. This study also suggests role of NRF2 mediated altered metabolism and stress granules for the same. Authors should add few lines on this topic as another possible approach for further research.

Thank you for pointing out this interesting topic. We have added a short paragraph about this starting on line 329: “Glutamine metabolism has also been suggested as a potential therapeutic target in KRAS mutant PDAC. In this context, mutant KRAS upregulates the transcription factor NRF2 which confers chemoresistance by altering pathways regulated by Gln metabolism. Glutamine depletion and use of glutaminase inhibitors sensitises PDAC cells to the chemotherapeutic agent gemcitabine[77]. Altered metabolic pathways offer another possible target for combination therapy in resistant tumours.”
However, we were unable find mention of NRF2 (or NFE2L2) in the manuscript by Hallin et al regarding MRTX849 resistance (DOI: 10.1158/2159-8290.CD-19-1167).

  1. Following up with above mentioned topic, it is also worthwhile mention about how to exploit metabolic checkpoints to break the resistance of KRAS G12C inhibitors. It has been mentioned in this manuscript elaborating the study from Lito's group stating that cell cycle arrest upon KRAS G12C inhibitors treatment. Lito's group has shown that treatment with G12Ci induced a quiescent state (G0) that was transcriptionally distinct from G1. While it has been shown (DOI:10.4161/23723548.2014.963445) that how to utilize metabolic checkpoints to create synthetic lethal phenotype in KRAS-mutants. Authors should add few lines on this aspect by referring mentioned works above. This will add a new aspect in this review which will help readers to think in broader aspect.

Thank you for this comment which we agree adds a new aspect to the “maximising therapeutic potential” section. We have added a new paragraph “cell cycle checkpoint co-inhibition” starting on line 312, including the following: “Metabolic reprogramming is an important hallmark of cancer cells and presents a potential method for targeting KRAS driven malignancies. In order to meet increased requirements for cell growth, cancer cells redirect intermediates from the tricarboxylic acid (TCA) cycle, which need to be replenished with the amino acid glutamine (Gln). The reliance on Gln creates a Gln dependent G1 cell cycle checkpoint, which KRAS mutant cancer cells are able to bypass in states of Gln depletion, instead arresting in S/ G2[75]. This can be exploited by utilising methods to induce Gln depletion, while simultaneously employing G2/S specific cytotoxic drugs [76]. This approach proves that metabolic checkpoints can be used in synthetic lethality screens. This approach could be used to overcome resistance to G12C inhibitors, which are known to sequester cells into a quiescent state, while adapting cells progress through G1/S and G2 [57].”

  1. in case of SHP2+KRAS G12C combination study, another new work came out from Novartis group (DOI: 10.1158/1078-0432.CCR-20-2718). It has been shown that SHP2 inhibitor TNO155 is efficient in KRAS G12C cancer cells: it efficiently blocked the feedback activation of wild-type KRAS or other RAS isoforms induced by G12C inhibitors and efficiently enhanced the efficacy of G12C targeted therapy.  

Thank you for this update. We have added this on line 253 as follows: “In agreement with these findings, the SHP2 inhibitor TNO155 has been shown to overcome feedback reactivation of RTK signalling induced by G12C inhibition, and was found to act synergistically with the G12C inhibitor Cpd 12a to inhibit proliferation in KRASG12C cell lines. This synergistic effect was the most significant of all TNO155 combinations tested in the study, and was also postulated to be secondary to increased GDP occupancy of KRAS[66].”

Round 2

Reviewer 2 Report

All concerns have been addressed. Ready for acceptance.